# Study protocol for a pragmatic randomised controlled trial comparing the effectiveness and cost-effectiveness of levetiracetam and zonisamide versus standard treatments for epilepsy: a comparison of standard and new antiepileptic drugs (SANAD-II)

Silviya Balabanova [ID],[1] Claire Taylor,[1] Graeme Sills,[2] Girvan Burnside,[3] Catrin Plumpton,[4] Phil E M Smith [ID],[5] Richard Appleton,[6] John Paul Leach,[7] Michael Johnson,[8] Gus Baker,[9] Munir Pirmohamed,[10] Dyfrig A Hughes,[4] Paula R Williamson,[11] Catrin Tudur-Smith,[3] Anthony Guy Marson[12]

For numbered affiliations see end of article.

**Correspondence to**
Professor Anthony Guy Marson; marjon01@liverpool.ac.uk

## ABSTRACT

**Introduction** Antiepileptic drugs (AEDs) are the mainstay of epilepsy treatment. Over the past 20 years, a number of new drugs have been approved for National Health Service (NHS) use on the basis of information from short-term trials that demonstrate efficacy. These trials do not provide information about the longer term outcomes, which inform treatment policy. This trial will assess the long-term clinical and cost-effectiveness of the newer treatment levetiracetam and zonisamide.

**Methods and analysis** This is a phase IV, multicentre, open-label, randomised, controlled clinical trial comparing new and standard treatments for patients with newly diagnosed epilepsy. Arm A of the trial randomised 990 patients with focal epilepsy to standard AED lamotrigine or new AED levetiracetam or zonisamide. Arm B randomised 520 patients with generalised epilepsy to standard AED sodium valproate or new AED levetiracetam. Patients are recruited from UK NHS outpatient epilepsy, general neurology and paediatric clinics. Included patients are aged 5 years or older with two or more spontaneous seizures requiring AED monotherapy, who are not previously treated with AEDs. Patients are followed up for a minimum of 2 years. The primary outcome is time to 12-month remission from seizures. Secondary outcomes include time to treatment failure (including due to inadequate seizure control or unacceptable adverse reactions); time to first seizure; time to 24-month remission; adverse reactions and quality of life. All primary analyses will be on an intention to treat basis. Separate analyses will be undertaken for each arm. Health economic analysis will be conducted from the perspective of the NHS to assess the cost-effectiveness of each AED.

**Ethics and dissemination** This trial has been approved by the North West-Liverpool East REC (Ref. 12/NW/0361). The trial team will disseminate the results through scientific meetings, peer-reviewed publications and patient and public involvement.

**Trial registration numbers** EudraCT 2012-001884-64; ISRCTN30294119.

### Strengths and limitations of this study

► The study is adequately powered to examine the clinical effectiveness of standard and new antiepileptic drugs (AEDs).
► The study is adequately powered to examine the cost-effectiveness of standard and AEDs.
► This is a long-term trial that will provide evidence of the long-term effects of some AEDs.
► A limitation is that the study is not blinded.
► A further limitation is the reliance on patient completed questionnaires, which should be offset by access to hospital episode statistic data.

## INTRODUCTION

Epilepsy is a common neurological condition and up to 3% of people will experience seizures at some time in their lives.[1] Epilepsy is a complex condition with many different causes and seizures can take many different forms. It is uniquely stigmatising and has a negative impact on quality of life (QoL) and employment prospects.[2 3] Antiepileptic drugs (AEDs) are the mainstay of treatment and may have to be taken for life. The ultimate goal of treatment is to maximise QoL by eliminating seizures at drug doses that do not cause side effects. However, for many patients, there is a

necessary trade-off between effective seizure control and side effects, which can diminish QoL.

Over the past 20 years, a number of new drugs have become available for the treatment of epilepsy. These new drugs have been approved for National Health Service (NHS) use on the basis of information from short-term trials. These trials do not provide information about the longer term outcomes, which inform decisions made by doctors and patients, nor do they provide any useful health economic data to inform policy.

The standard and new antiepileptic drugs (SANAD-I) trial began in 1999 and compared the effectiveness and cost-effectiveness of standard and new treatments that were available at that time.[4 5] SANAD-I identified lamotrigine (a new drug) as an effective and cost-effective first-line treatment for patients with a focal epilepsy and confirmed that valproate (a standard treatment) should remain a first-line drug for patients with a generalised epilepsy or seizures that clinicians find difficult to classify. Since SANAD-I, a number of newer treatments have become available, the most promising of that are levetiracetam and zonisamide. SANAD-II will include two randomised controlled trials (RCTs) run in parallel; arm A recruiting patients with focal epilepsy and arm B patients with generalised or unclassified epilepsy.

## Rationale
### Arm A
Arm A (focal epilepsy) of SANAD-II will compare lamotrigine, levetiracetam and zonisamide in patients with untreated focal onset seizures. While the focal epilepsies are further classified into a number of syndromes[6] largely according to aetiology and site of onset, it has been common practice to recruit a heterogeneous population with focal onset seizures into epilepsy trials. There are currently no reliable data that indicate whether relative treatment responses differ among the focal epilepsies, indeed prognostic modelling of data from SANAD-I suggests that treatment effects are consistent across focal epilepsy syndromes as currently classified.[4 7] In SANAD-II, we will recruit patients with focal onset seizures irrespective of syndrome, which will allow further opportunity to investigate treatment effects and factors, including syndrome, that influence those effects.

Lamotrigine was chosen as the 'standard' treatment as in SANAD-I, it was found to be superior to carbamazepine for time to treatment failure and non-inferior to carbamazepine for time to 12-month remission, while the health economic analysis found lamotrigine a cost-effective alternative to carbamazepine.[4]

Levetiracetam is a commonly used AED with evidence for efficacy (non-inferiority (NI) to carbamazepine for 6-month seizure remission) as monotherapy in focal epilepsy from regulatory studies with too short a duration of follow-up to inform policy.[8] A second industry-sponsored unblinded trial compared levetiracetam with physicians' choice of carbamazepine or valproate, but again follow-up was too short to inform treatment policy.[9]

Zonisamide is a drug that has been available for many years in Japan and other countries in South East Asia where it is commonly used both as initial monotherapy and as an add-on treatment and is licensed for use in the European Union and USA. Evidence for efficacy is from industry-sponsored regulatory studies demonstrating NI when compared with carbamazepine for 6-month seizure remission rates.[10]

### Arm B
Arm B (generalised or unclassified epilepsy) of SANAD-II will compare levetiracetam and valproate in patients with generalised onset seizures or seizures that are difficult to classify. Generalised onset seizures represent a group of syndromes, most of which are currently classified as one of the idiopathic generalised epilepsies,[6] which are largely classified according to seizure type and age of onset. While differing syndromes are recognised, there is currently no reliable evidence that relative treatment responses differ among syndromes, indeed prognostic modelling of data from SANAD-I indicates that relative treatment responses are consistent across syndromes. As in SANAD-I, patients enter Arm B of the trial based on a classification of seizures (generalised onset or difficult to classify), with patients further classified by syndrome where and when such a syndromic diagnosis can be made.

Few RCTs have been undertaken to assess the comparative effects of AEDs in patients with generalised onset seizures or in those with seizures who are difficult to classify, even though these individuals represent over one-third of people with epilepsy. Valproate has for some time been recommended as a first-line treatment for such patients[11] but without evidence from RCTs to support this recommendation. Cochrane reviews have compared valproate with other AEDs,[12 13] but due to problems with power and epilepsy classification, none has shown an advantage for valproate. In arm B of SANAD-I,[5] valproate was significantly more effective than lamotrigine and significantly better tolerated than topiramate. Also, a double-blind trial of 16 weeks therapy in childhood and juvenile absence epilepsy,[14] valproate and ethosuximide were significantly superior to lamotrigine for the outcome treatment failure.

Valproate, however, remains a difficult drug for women of childbearing potential as it is associated with a higher rate of teratogenicity than alternatives (major malformation rate ~8%).[15] There is also evidence that valproate can affect the intellectual development of children exposed in utero with up to one-third of children having a significant reduction in their IQ.[16] In 2018, the European Medicines Agency (EMA) and Medicines and Healthcare products Regulatory Authority (MHRA) launched a pregnancy prevention programme for women taking valproate, and state that women of childbearing potential should not be prescribed valproate unless other treatments are ineffective or not tolerated.[17] For women with generalised epilepsy, making a treatment choice is very challenging, with options including valproate, lamotrigine, which

is less effective but safer in pregnancy or levetiracetam for which the efficacy when compared with valproate is unknown and for which we have increasing evidence of relative safety in pregnancy.

## TRIAL OUTCOMES
### Primary outcome
The primary outcome of the trial is time to 12-month remission from seizures.

### Secondary outcomes
The secondary outcomes are
1. Time to treatment failure.
2. Time to treatment failure due to inadequate seizure control. This event will have occurred when the clinician and/or patient decides that treatment replacement or withdrawal or the addition of a second AED is required due to the occurrence of a seizure on the maximum recommended dose of randomised drug or the maximum tolerated dose of the drug.
3. Time to treatment failure due to unacceptable adverse events. This event will have occurred when the patient experiences adverse events attributed to the drug necessitating its withdrawal.
4. Time to first seizure.
5. Time to 24-month remission.
6. Adverse reactions (ARs).
7. QoL.
8. Health economic outcomes expressed as the incremental cost per quality-adjusted life-year (QALY) gained.

## TRIAL DESIGN
SANAD-II trial is a phase IV, multicentre, randomised, open-label controlled trial. Arm A of the trial recruits participants diagnosed with focal epilepsy and arm B—with generalised or unclassified epilepsy. Trial participants are randomised to an AED within each arm. In arm A, the control treatment is lamotrigine and the new treatments are levetiracetam and zonisamide, in treatment group ratio of 1:1:1. In arm B, the new treatment levetiracetam is compared with the control treatment valproate in treatment group ratio of 1:1. The sample size for the trial is 1510 participants (990 with focal epilepsy and 520 with generalised or unclassified epilepsy).

### Enrolment and randomisation
#### Screening and baseline
The SANAD-II trial will take place in NHS outpatient epilepsy, general neurology and paediatric (epilepsy and general) clinics in the UK (a full list of participating sites is available from the trial website). All patients meeting the eligibility criteria (table 1) will be invited to participate in the study and provided with an age and development-appropriate patient information sheet and consent form. Patients will be allowed sufficient time to discuss trial and to decide on their participation. Once consent has been

**Table 1** Eligibility criteria

| Inclusion criteria | a. Aged 5 years or older. |
| --- | --- |
| | b. Two or more spontaneous seizures that require antiepileptic drug treatment. |
| | c. Untreated and not previously treated with antiepileptic drugs, except emergency treatment in the past 2 weeks. |
| | d. Antiepileptic drug monotherapy considered the most appropriate option. |
| | e. Willing to provide consent (patients' parent/legal representative willing to give consent where the patient is aged under 16 years or is lacking capacity to consent). |
| Exclusion criteria | a. Provoked seizures only (eg, alcohol or drug induced). |
| | b. Acute symptomatic seizures only (eg, within 1 month from acute brain haemorrhage or brain injury or stroke). |
| | c. Currently treated with antiepileptic drugs. |
| | d. Progressive neurological disease (eg, known brain tumour). |

obtained, the baseline data will be collected and patient will be progressed to randomisation.

### Randomisation
The arm (A or B) to which the patient is assigned will be decided by the recruiting physician based on their epilepsy classification. Patients will then be randomised to one of the following treatments: to lamotrigine, levetiracetam or zonisamide (in arm A) or to levetiracetam or valproate (in arm B). Randomisation will use a minimisation programme with a built-in random element using factors that will not be made known to individuals in charge of recruitment to minimise any potential for predicting allocation. Participants will be randomised using a secure (24 hours) web-based randomisation programme controlled centrally by the Clinical Trials Research Centre (CTRC). Randomised treatment should begin within 7 days of randomisation. The research team should ensure that the duration between obtaining consent, performing baseline assessments, randomisation and the start of trial treatment does not impact on the well-being of the participant.

### Trial interventions
SANAD-II is a pragmatic trial that uses market authorised drugs within the terms of marketing authorisation. All treatments will be taken as formulations already licensed to be used in UK and there will be no modifications made to the products or their outer packaging. All treatments will be prescribed as per routine NHS practice and dispensed by hospital and community pharmacies as they would normally be.

It is accepted that, for a variety of reasons including perceived or actual efficacy and tolerability, not all

patients will take their medicines as prescribed. Patients will be asked about adherence in the QoL questionnaires, but no objective measurements of adherence are planned nor will the primary analyses be adjusted for actual or estimated adherence.

SANAD-II is an unblinded trial therefore decisions about concomitant medications/treatments will depend on the local medical plan and clinical management.

All patients will be titrated to an initial maintenance dose, with dose adjustments made at subsequent appointments according to the clinical response and adverse effects. Guidelines for titration and initial maintenance dose are provided within the protocol (table 2), however, clinicians will be able to alter this to choose the titration rate and initial maintenance they think most appropriate for individual patients.

**Table 2**  Titration and initial maintenance dose

**Arm A**

**Age >12 years**

| Lamotrigine | Levetiracetam | Zonisamide |
|---|---|---|
| 25 mg once per day for 2 weeks | 250 mg once per day for 2 weeks | 50 mg once per day 2 weeks |
| 25 mg two times per day for 2 weeks | 250 mg two times per day for 2 weeks | 50 mg two times per day for 2 weeks |
| 50 mg two times per day for 2 weeks | 250 mg morning and 500 mg night for 2 weeks | 50 mg am and 100 mg pm for 2 weeks |
| 50 mg morning and 100 mg at night—*initial target maintenance dose* | 500 mg two times per day—*initial target maintenance dose* | 100 mg am 100 mg pm—*initial target maintenance dose* |

**Children aged 5–12 years**

| Lamotrigine | Levetiracetam | Zonisamide |
|---|---|---|
| 0.5 mg/kg/ day as once a day dose for 2 weeks | 10 mg/kg/day as two times daily regimen for 2 weeks | 0.5–1 mg/kg/day as once or two times daily regimen (depending on the child's weight) for 2 weeks |
| 0.5 mg/kg/day as two times daily regimen for 2 weeks | 20 mg/kg/day as two times daily regimen for 2 weeks | 1–1.5 (maximum) mg/kg/day as two times daily regimen for 2 weeks |
| 0.5 mg/kg am and 1.0 mg/kg pm for 2 weeks | 30 mg/kg/day as two times daily regimen for 2 weeks | 2–2.5 (maximum) mg/kg/day as two times daily regimen for 2 weeks |
| 1.0 mg/kg am and 1.0 mg/kg pm for 2 weeks | 40 mg/kg/day as two times daily regimen—*initial target maintenance dose* | 3–4 mg/kg/day as two times daily regimen for 2 weeks |
| 1.5 mg/kg am and 1.5 mg/kg pm—*initial target maintenance dose* | | 5 mg/kg/day as two times daily regimen—*initial target maintenance dose*. |

**Arm B**

**Age >12 years**

| Valproate | Levetiracetam |
|---|---|
| 500 mg once per day for 2 weeks | 250 mg once per day for 2 weeks |
| 500 mg two times per day—*initial target maintenance dose* | 250 mg two times per day for 2 weeks |
| | 250 mg morning and 500 mg night for 2 weeks |
| | 500 mg two times per day—*initial target maintenance dose* |

**Children aged 5–12 years**

| Valproate | Levetiracetam |
|---|---|
| 10 mg/kg/day as two times daily regimen for 2 weeks | 10 mg/kg/day as two times daily regimen for 2 weeks |
| 15 mg/kg/day as two times daily regimen for 2 weeks | 20 mg/kg/day as twotimes daily regimen for 2 weeks |
| 25 mg/kg/day as two times daily regimen—*initial target maintenance dose* | 30 mg/kg/day as two times daily regimen for 2 weeks |
| | 40 mg/kg/day as two times daily regimen—*initial target maintenance dose* |

**Table 3** Schedule of follow-up

| Procedures | Baseline (T0)* | Follow-up schedule | | |
|---|---|---|---|---|
| | | T0+3 months | T0+6 months | T0+12 months and annually thereafter |
| Signed consent form | X | | | |
| Assessment of eligibility criteria | X | | | |
| Contact details | X | | | |
| Review of medical history including: | X | | | |
| ► Seizure history | | | | |
| ► Neurological insult | | | | |
| ► Febrile seizures | | | | |
| ► Family history of epilepsy | | | | |
| ► EEG results | | | | |
| ► Imaging results (CT or MRI) | | | | |
| Further investigation (EEG/CT/MRI) | (X) | | | |
| Allocation of study treatment | X | | | |
| Issue of questionnaires in person or by post | X | X | X | X |
| Review of seizure occurrence and hospital admissions | | X | X | X |
| Review of AED use (study treatment and concomitant): | | | | |
| ► Since last follow-up | | X | X | X |
| ► Changes made to treatment plan including reasons | | | | |
| Assessment of adverse reactions | | (X) | (X) | (X) |
| Resource use | | X | X | X |
| Reissue of questionnaire by post or at site to non-responders typically 3 weeks later | (X) | (X) | (X) | (X) |
| Telephone follow-up of questionnaire non-responders typically 3 weeks later | (X) | (X) | (X) | (X) |
| Special assay or procedure consent and obtain saliva or blood sample for later DNA analysis | Consent and obtain saliva or blood sample for later DNA analysis (X) | | | |

(X)—as indicated/appropriate.
EEG = Electroencephalogram
*At baseline, all procedures should be done before study intervention.
AED, antiepileptic drug.

The aim of treatment will be to control seizures with a minimum effective dose of drug. This will necessitate dosage modification (dose increased or reduced) if further seizures or adverse events occur as is usual clinical practice. Any changes in medication must be documented along with the justification for those changes. At the end of trial participation, the participants may continue their treatment as per local policy.

To avoid potentially confounding issues, ideally patients should not be recruited into other epilepsy trials.

### Assessments and procedures
#### Schedule for follow-up
Patients were recruited over a 4.5-year period and follow-up will continue for further 2 years. The maximum time that a patient will receive their randomised treatment is 6.5 years. Table 3 shows the schedule of follow-up.

All participants will be followed up whether they are still taking their allocated treatment or not. Where patients default from clinic follow-up, additional information will be sought from general practitioners who will be the main prescribers of AEDs in this trial. Patients will be followed up as per routine clinical practice and typically at 3, 6 and 12 months and annually thereafter. Patients may be seen at other times as clinically indicated.

Where treatment is stopped, the participant will be asked to continue with the trial follow-up and to attend the follow-up visits.

Efficacy of the trial treatments will be measured throughout the trial using a number of measures. Data on seizures will provide a subjective measure of efficacy. QoL data obtained throughout the trial using age-appropriate

**Table 4** Age-appropriate questionnaire booklets

| Participant age | Questionnaire booklet completed by | Questionnaire booklet content |
|---|---|---|
| 5–7 | Participant | N/A |
| | Parent/carer | Kiddy-KINDL, EQ-5D-3L & EQ-VAS, NEWQOL-6D |
| 8–11 (children) | Participant | Kid-KINDL, EQ-5D-3L-Y & EQ-VAS, QOLIE-AD |
| | Parent/carer | Kid-KINDL, EQ-5D-3L & EQ-VAS, NEWQOL-6D |
| 12–15 (young people) | Participant | Kiddo-KINDL, EQ-5D-3L-Y & EQ-VAS, QOLIE-AD |
| | Parent/carer | Kiddo-KINDL, EQ-5D-3L & EQ-VAS, NEWQOL-6D |
| ≥16 (adult) | Participant | Impact of Epilepsy Scale, EQ-5D-3L & EQ-VAS, NEWQOL-6D |

questionnaire booklets (table 4) can be used as a subjective measure of efficacy.

Assessment of ARs will be undertaken at each study visit.

## QOL and utility assessments

For adults, QOL outcomes will be assessed using subscales of the Quality of Life in Newly Diagnosed Epilespy (NEWQOL) battery and the Impact of Epilepsy Scale.[18] For children and adolescents aged <16 years, QoL assessment will involve both patient and parent-based measures: children aged 8–15 years will complete a generic health status measure validated for use in epilepsy, the KINDL[19]; and the 'epilepsy impact' and 'attitude to epilepsy' subscales of the Quality of Life in Epilepsy for Adults (QOLIE-AD).[20] Parents of all children will also complete proxy QoL questionnaires.

Utility scores will be elicited directly from trial participants (or indirectly via parents/guardians). Adult and adolescent patients will be asked to complete the EQ-5D-3L questionnaire and Visual Analogue Scale. The EQ-5D-3L has been used previously in children, but it has not been formally validated,[21] and EQ-5D-3L weights are validated for adults aged ≥18 years. The currently recommended approach of using parental proxy reports of QoL for this age group will be used.[22] EQ-5D-3L-Y (youth version) will additionally be administered to children aged 8–15 years. All trial participants will be asked to complete an epilepsy-specific utility measure, based on the NEWQOL-6D questionnaire.[23]

Patients and/or parents/guardians will be completing questionnaires at baseline and at specific time points throughout the trial.

## Resources use and cost data

Direct costs of healthcare resources used by patients in the trial will be collected in three ways:

1. A modified version of the Client Service Receipt Inventory[24] to assess patients' use of primary and community care services and personal social services (eg, primary care services (NHS Direct, walk-in treatment centres, home visits, etc).
2. Patients' use of secondary care services as Hospital Episode Statistics (HES) data. Downloaded Healthcare Resource Group data will include information on outpatient epilepsy, general neurology and paediatric clinics visits; accident and emergency attendance and length (and nature) of hospitalisations.
3. Resources triggered by ARs will be captured in the follow-up case report form (CRF) for each patient experiencing a serious AR requiring hospitalisation. Because of potential issues related to completeness of routine data, these will be used to compliment HES data.
4. Unit costs will be taken from the NHS reference costs database[25] and other appropriate sources.[26 27]

## Genetic substudy

DNA collection will be included as an option in SANAD-II in order that refusal will not preclude from trial participation. Whole blood or saliva samples will be shipped to a central laboratory at The University of Liverpool for extraction and storage. No genotyping of DNA samples will be undertaken as part of SANAD-II. Samples will instead be genotyped in future projects and data arising from that analysis included in international epilepsy genomics initiatives. Identification of genetic factors associated with response to treatment in epilepsy is important and may ultimately help to optimise efficacy, tolerability and safety of AEDs.

## Sample size calculation

SANAD-II is powered to detect NI of the new AEDs (levetiracetam and zonisamide) compared with standard treatments (lamotrigine or valproate) for the primary outcome time to 12-month remission. A new drug might become a standard first-line treatment if it is proven to be non-inferior for efficacy but superior for tolerability when compared with a standard treatment—tolerability is examined in secondary outcomes including time to treatment failure for adverse effects. Powering the study for NI will also provide sufficient power to detect important differences between treatment policies.

The International League Against Epilepsy (ILAE) Commission on Antiepileptic Drugs defined limits of equivalence of ±10% for the primary outcome in AED monotherapy studies.[28] However, the commission was not explicit as to whether this should be on the HR or absolute scale. No empirical work has yet been undertaken

to underpin the choice of equivalence or NI margins in epilepsy trials. The chief investigator has given numerous seminars and lectures in the UK and elsewhere about epilepsy trial methodology and the audience typically vote for a margin of 10% around absolute differences between AEDs for monotherapy studies when given examples of margins ranging from 20% to 5%.

Calculations have been informed by the SANAD-I study, which estimated the 12-month remission-free probability (at 24 months) as 0.43 (exponential hazard rate of 0.0352) for lamotrigine (arm A standard) and 0.31 (exponential hazard rate of 0.0488) for valproate (arm B standard). The calculations assume a HR of 1.0, 80% power, and allowance for approximately 5% losses to follow-up throughout, as occurred in SANAD-I. For patients with focal-onset seizures (arm A), two primary comparisons are of interest (levetiracetam vs lamotrigine and zonisamide vs lamotrigine), therefore, the one-sided significance level has been divided by two (one-side alpha 0.0125). Assuming a 10% absolute difference in survival probability, the NI margin on the HR scale is $\ln(0.43)/\ln(0.53)=1.329$. After adjusting for 5% losses to follow-up, 330 patients are required in each of three treatment groups (990 total for arm A). For patients with generalised-onset seizures or seizures that are difficult to classify (arm B), there is only one comparison of interest (levetiracetam vs valproate). Assuming a 10% absolute difference in survival probability, the NI margin on the HR scale is $\ln(0.31)/\ln(0.41)=1.314$ for arm B. Therefore, with a one-sided alpha of 0.025, 260 patients are required in each of two treatment groups allowing for 5% losses to follow-up (520 total for arm B). The total number of patients required is 1510.

## Statistical analysis

All primary analyses will be on an intention to treat (ITT) basis including all randomised patients retained in their randomised treatment groups. Separate analyses will be undertaken for each randomisation arm. The interval (in days) from randomisation to occurrence of a 12-month remission will be summarised by Kaplan-Meier curves for each treatment group. Survival regression models will be explored; two different models will be used: (1) including the treatment effect only using treatment indicator variables and (2) including the treatment effect together with covariates. The impact of centre effect on the treatment comparison will be investigated using both fixed and random effect models. A per protocol analysis will be undertaken to assess the robustness of ITT analyses. For arm B trial, an additional analysis of the primary outcome will add a stratification variable to the model: seizure type (generalised/unclassified).

A similar analysis strategy will be employed for the other secondary time to event outcomes. For time to treatment failure, further analysis will be undertaken to assess the two main reasons for treatment failure—inadequate seizure control and unacceptable adverse effects. To allow for possible dependence between the different withdrawal risks, cumulative incidence analyses will be presented.[29]

The Haybittle-Peto approach will be employed for each interim analysis, with 99.9% CIs calculated for interim analysis effect estimates. The final analysis will be undertaken at the end of the trial when all patients have a minimum 2-year follow-up data (6.5 years after the first patient is randomised) and 95% CIs will be calculated.

QoL data will be analysed longitudinally to explore between treatment changes in scale scores over time, taking account of baseline QoL.

For the analysis of ARs, all patients who received any amount of each study drug will be included in the safety analysis dataset in the treatment group they actually received. All ARs and serious ARs reported by the clinical investigators will be presented, identified by treatment group. ARs will be grouped according to a prespecified coding system and tabulated. The number (and percentage) of patients experiencing each AR, and the number (and percentage) of occurrences of each AR will be presented. No formal statistical testing will be undertaken.

## Health economic evaluation

For the health economic analyses, the perspective of the NHS and Personal Social Services will be adopted for costing purposes. It will be assessed whether levetiracetam or zonisamide as monotherapy in newly treated focal epilepsy is cost-effective by estimating the incremental cost-utility and cost-effectiveness ratios relative to lamotrigine and to each other. The same approach will be used to compare levetiracetam and valproate for generalised-onset seizures. A cost consequence analysis[30] will be conducted to consider non-health benefits that are neither captured within the QALY calculation, nor in the cost-effectiveness analyses. Potential non-health benefits that will be measured include social activity, time in work or school and patients' driving (captured in the NEWQOL battery). Additional non-health benefits (perceived stigma, control and cognitive effects) will also be captured in the NEWQOL-6D.[23]

Sensitivity analyses will be conducted to test the robustness of our findings. These analyses will be based on the observed distributions of outcome and costs to test whether, and to what extent, the incremental cost-utility and cost-effectiveness ratios are sensitive to key assumptions in the analysis. Uncertainty in parameter estimates will be addressed through the application of bootstrapping and the estimation of cost-effectiveness acceptability curves.

The estimated incremental cost per QALY will be compared with the threshold for cost-effectiveness operating in the UK, and the incremental cost per seizure will be avoided and per 12-month remission will be compared with the results of other economic assessments of AEDs.[31 32]

## Ethics and dissemination

### Ethics

The trial protocol complies with the Standard Protocol Items: Recommendations for Interventional Trials reporting guidelines.[33] All relevant trial documentations have been approved by the North West—Liverpool East Research Ethics Committee (Ref: 12/NW/0361). This trial falls within the remit of the EU Directive 2001/20/EC,[34] transposed into UK law as the UK Statutory Instrument 2004 No 1031: Medicines for Human Use (Clinical Trials) Regulations 2004 as amended.[35] SANAD-II has been registered with the MHRA and was granted a Clinical Trial Authorisation for Notification prior to initiation.

### Dissemination

The trial team plans to disseminate the SANAD-II results by presentations at national (eg, Association of British Neurologists (ABN) and international (eg, ILAE—European congress, or ILAE international Epilepsy Congress) meetings as well as publications in leading peer-reviewed journals and through patient and public involvement. Authorship of any documents will follow International Committeee of Medical Journal Editors (ICMJE) Recommendations 2018.

## Data collection and trial monitoring

### Data collection

Data management for SANAD-II is performed by the CTRC at University of Liverpool. Participating centres will be expected to each maintain a file of essential trial documentation (Site File), which will be provided by the coordinating centre and keep copies of all completed CRFs. Data collection will use paper CRFs and participant completed questionnaires.

### Trial monitoring

Trial Oversight Committees have been formed in relation to the monitoring. This includes an Independent Data and Safety Monitoring Committee (IDSMC), consisting of independent epilepsy experts and statisticians; A Trial Management Group (TMG) consisting of the chief investigator, trial manager, trial statisticians, sponsor representatives, several principal investigators and a Trial Steering Committee (TSC) consisting of two independent epilepsy clinicians, an independent statistician and a lay representative, in addition to select members of the TMG.

Trial monitoring is informed by a risk assessment to determine the level and type of monitoring required for specific hazards. The IDSMC will monitor participant safety including any unexpected ARs. ARs will be reported to the MHRA and to the Liverpool East Research Ethics Committee via annual Data and Safety Monitoring Reports. The committees will also monitor recruitment rates and adherence to Good Clinical Practice (GCP) guidance.

### Patient and public involvement

SANAD-II was designed in collaboration with epilepsy action and has a patient and public involvement representative in the TSC.

### Time frame and trial status

The study opened to recruitment in 2013 and completed recruitment in June 2017. Participants are followed up for a minimum of 2 years and a maximum of 6.5 years. Follow-up visits and data collection will continue until June 2019.

**Author affiliations**
¹Liverpool Clinical Trials Centre, University of Liverpool, Faculty of Health and Life Sciences, Liverpool, UK
²School of Life Sciences, University of Glasgow, Glasgow, UK
³Biostatistics, University of Liverpool, Faculty of Health and Life Sciences, Liverpool, UK
⁴Centre for Health Economics and Medicines Evaluation, Bangor University, Bangor, UK
⁵Department of Neurology, University Hospital of Wales, Cardiff, UK
⁶Paediatric Neurology, Alder Hey Children's NHS Foundation Trust, Liverpool, UK
⁷School of Medicine, University of Glasgow, Glasgow, UK
⁸Department of Brain Sciences, Imperial College London Faculty of Medicine—South Kensington Campus, London, UK
⁹Molecular and Clinical Pharmacology, University of Liverpool, Faculty of Health and Life Sciences, Liverpool, UK
¹⁰Department of Pharmacology, University of Liverpool Faculty of Health and Life Sciences, Liverpool, UK
¹¹Biostatistics, University of Liverpool, Liverpool, UK
¹²Molecular and Clinical Pharmacology, University of Liverpool, Liverpool, UK

**Acknowledgements** We would like to acknowledge the input of the wider SANAD-II team and to thank the research sites and participants for being involved in this study. We would like to specifically thank Epilepsy Action for their invaluable contribution throughout the trial.

**Contributors** GS, PEMS, RA, JPL, MJ, GBa, MP, DAH, PRW, CT-S and AGM were involved in the trial design and the grant application preparation. SB, GS, GBu, CP, PEMS, RA, JPL, MJ, GBa, MP, DAH, PRW, CT-S and AGM were involved in the protocol development and protocol amendments throughout the trial. SB, CT and AGM were responsible for the drafting of the manuscript. All authors were able to contribute to the draft and approved the final text.

**Funding** This work is supported by the Health Technology Assessment Programme (HTA) of National Institute for Health Research (NIHR) grant number 09/144/09. The SANADII trial is cosponsored by the University of Liverpool (Sponsor Representative Alex Astor [sponsor@liverpool.ac.uk]) and The Walton Centre (Sponsor Representative Martin Wilby [RDI@thewaltoncentre.nhs.uk]).

**Competing interests** None declared.

**Patient and public involvement** Patients and/or the public were involved in the design, or conduct, or reporting, or dissemination plans of this research. Refer to the Methods section for further details.

**Patient consent for publication** Not required.

**Provenance and peer review** Not commissioned; externally peer reviewed.

**ORCID iDs**
Silviya Balabanova http://orcid.org/0000-0001-6989-8099
Phil E M Smith http://orcid.org/0000-0003-4250-2562

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
