## [Reviewer comments · BMJ Open]

ARTICLE DETAILS

TITLE (PROVISIONAL)	A study protocol for a pragmatic randomised controlled trial comparing the effectiveness and cost effectiveness of levetiracetam and zonisamide versus standard treatments for epilepsy: a comparison of Standard And New Antiepileptic Drugs (SANAD-II).
AUTHORS	Balabanova, Silviya; Taylor, Claire; Sills, Graeme; Burnside, Girvan; Plumpton, Catrin; Smith, Phil; Appleton, Richard; Leach, John; Johnson, Michael; Baker, Gus; Pirmohamed, Munir; Hughes, Dyfrig; Williamson, Paula; Tudur-Smith, Catrin; Marson, Anthony

VERSION 1 – REVIEW

REVIEWER	rani sarkis Brigham and Women's Hospital, Harvard medical school, USA
REVIEW RETURNED	29-Jul-2019

GENERAL COMMENTS	Balabanova et al. describe their Protocol for SANAD-II a pragmatic trial looking at : Levetiracetam v.s. Lamotrigine v.s. Zonisamide in focal epilepsy and Levetiracetam v.s. Valproic acid in Generalized or unclassified epilepsy. The SANAD trials have been essential in informing epilepsy clinical practice worldwide and an update on SANAD-1 is needed given that Zonisamide (ZNS) and levetiracetam (LEV) are commonly prescribed. The protocol is will designed. I have 2 comments about to the protocol 1- My main concern is that the lamotrigine group seems under-dosed at a target of 150mg a day without any measurement of levels. Given that the outcome is time to remission, I would predict that the LEV and ZNS group will be therapeutic at the end of the titration and the lamotrigine arm will not, thus impacting outcomes. In clinical practice most target doses are 200mg and the WHO defined daily dose is 300mg. The authors should explain the rationale for such an approach. 2- The authors should plan a subset analysis of Arm B given that IGE patients and unclassified are lumped together. An IGE-only analysis where the physician is confident that the patient has generalized epilepsy is needed.
--

REVIEWER	Meritxell Martinez Ferri Universitary Hospital Mutua Terrassa Barcelona Spain
REVIEW RETURNED	07-Aug-2019

GENERAL COMMENTS	1) As in SANAD I with a carbamacepine initial dose not well tolerated, in this study initial maintenance dose of Valproate 1000
---

	mg /day and Levetiracetam 1000mg /day were not comparable in arm B; the same for Lamotrigine 150 mg and Zonisamide 200 mg/day in arm A on focal epilepsies. The titration to a higher than necessary initial maintenance dose of valproate or zonisamide might be biased in favor of less side effects of levetiracetam or lamotrigine or biased in favor of valproate for more seizure control. 2) as in SANAD I, interpretation of data from arm B is difficult because generalized epilepsies include many types of seizures . For example there is no evidence that levetiracetam has any efficacy for Absence seizures. To correct this fault the authors will make a subanalysis of the efficacy on the different types of seizures. This will be even more interesting in Arm B than the primary outcome " time to 12 month remission". 3) My other concern is about Women of child-bearing age. Valproate is contraindicated unless other treatments are ineffective or not tolerated. Levetiracetam has increasing evidence of its safety in pregnancy. (Tomson T et al. Lancet Neurol 2018). It is likely that few young women will choose to be randomized. So the comparative efficacy in this group of patients will remain unknown. 4) It will be also important to take attention to concomitant medication or to new concomitant medication. Patients could have breakdown seizures on LTG or valproate and less frequently on LEV or zonisamide with the introduction for example of oral contraceptives. Men and women could have different efficacies of AEDs some times related to hormonal states (Martinez Ferri M et al Neurologia 2006)
--	---

REVIEWER	Jacqueline French New York University. New York, NY USA I have a similar trial, the HEP study underway. I receive NYU salary support from the Epilepsy Foundation and for consulting work and/or attending Scientific Advisory Boards on behalf of the Epilepsy Study Consortium for UCB, Glaxo-SmithKline, and multiple other pharmaceutical companies for the purpose of drug development. I do not take independent money for this activity.
REVIEW RETURNED	08-Aug-2019

GENERAL COMMENTS	The protocol for the second study of SANAD is of great interest and importance to the epilepsy community, and the methodology as a whole is top-notch. This is a team that has honed this methodology through several prior collaborations. I have a few comments thought 1. The new ILAE definition of treatment response is as follows: a person's epilepsy can be classified as "drug responsive" ...if he/she has been seizure-free for a minimum of three times the longest pretreatment interseizure interval, or 12 months, whichever is longer. The study design and outcome does not incorporate this definition, in that the pre-treatment inter-seizure interval is not considered. Inclusion criteria are 2 prior seizures, and there is no time limit- Therefore a subject could have had 2 seizures 6 months or even a year apart, yet the outcome is time to 12 month seizure remission. At the very least, the period of time necessary to determine treatment response by the ILAE definition should be calculated for
--

	each subject, and should be provided in the outcome report. Ideally, the outcome should be time to treatment response as defined individually for each subject. 2. In the prior SANAD study, the seizure type was not taken into consideration. This is important for both focal and generalized epilepsy, and would be important for treatment providers. It would be very useful to include information of control of seizures with impaired awareness for the focal protocol, and control of GTCC for the generalized protocol. For example, if valproate and levetiracetam were equally as efficacious for treating GTCC but not myoclonus and absence, some patients might opt to use it if pregnancy was anticipated. 3. An ITT analysis is most rigorous in placebo-controlled trials, but not in comparative trials, because a subject who is quickly removed from a drug and placed on a different drug will be analysed in their original group, and side effects and seizure outcome will be attributed to the drug they were originally randomized to. This choice should be justified.
--	--

REVIEWER	Arif Khan, MD Northwest Clinical Research Center, Bellevue, WA and Duke University School of Medicine, USA
REVIEW RETURNED	23-Aug-2019

GENERAL COMMENTS	This is a complex one hundred pages of dense writing by a well grounded and experienced group seizure disorder specialists. Essentially, the paper is dedicated to the nature and methods to be used in a prospective trial assessing day to day clinical utility of two seizure medications. Much of the design is based on a similar trial conducted by this group more than a decade ago. Having said, my only concern is with the possibility that the minor changes such as the duration of dependent measure may increase the risk of a type II error. It is up to the editors of the journal as to how much trimming of the manuscript from the current 100 pages or so they want.
---

VERSION 1 – AUTHOR RESPONSE

Reviewer 1:

1- My main concern is that the lamotrigine group seems under-dosed at a target of 150mg a day without any measurement of levels. Given that the outcome is time to remission, I would predict that the LEV and ZNS group will be therapeutic at the end of the titration and the lamotrigine arm will not, thus impacting outcomes. In clinical practice most target doses are 200mg and the WHO defined daily dose is 300mg. The authors should explain the rationale for such an approach.

Trial team: We have followed NICE guidelines which suggest an initial maintenance dose of lamotrigine of 100 – 200mg. The protocol allows clinicians to choose the most appropriate dosage for individual patients.

2- The authors should plan a subset analysis of Arm B given that IGE patients and unclassified are lumped together. An IGE-only analysis where the physician is confident that the patient has generalized epilepsy is needed.

Trial team: This can be accommodated in the final analyses

Reviewer 2:

1) As in SANAD I with a carbamazepine initial dose not well tolerated, in this study initial maintenance dose of Valproate 1000 mg /day and Levetiracetam 1000mg /day were not comparable in arm B; the same for Lamotrigine 150 mg and Zonisamide 200 mg/day in arm A on focal epilepsies. The titration to a higher than necessary initial maintenance dose of valproate or zonisamide might be biased in favor to less side effects of levetiracetam or lamotrigine or biased in favor of valproate for more seizure control.

Trial team: We have followed NICE guidelines for dosages of all IMP. The protocol allows clinicians to choose the most appropriate dosage for individual patients, which should limit side effects.

2) as in SANAD I, interpretation of data from arm B is difficult because generalized epilepsies include many types of seizures. For example there is no evidence that levetiracetam has any efficacy for Absence seizures. To correct this fault the authors will make subanalysis of the efficacy on the different types of seizures. This will be even more interesting in Arm B than the primary outcome "time to 12 month remission".

Trial team: This can be accommodated

3) My other concern is about Women of child-bearing age. Valproate is contraindicated unless other treatments are ineffective or not tolerated. Levetiracetam has increasing evidence of its safety in pregnancy. (Tomson T et al. Lancet Neurol 2018). It is likely that few young women will choose to be randomized. So the comparative efficacy in this group of patients will remain unknown.

Trial team: An interesting point, and we were aware of the potential effects of valproate on pregnancy. Valproate was not contraindicated in women of childbearing age at the start of the study (2013)...

4) It will be also important to take attention to concomitant medication or to new concomitant medication. Patients could have breakdown seizures on LTG or valproate and less frequently on LEV or zonisamide with the introduction for example of oral contraceptives. Men and women could have different efficacies of AEDs some times related to hormonal states (Martinez Ferri M et al Neurologia 2006)

Trial team: This can be added to our analyses.

Reviewer 3:

1. The new ILAE definition of treatment response is as follows: a person's epilepsy can be classified as "drug responsive" ...if he/she has been seizure-free for a minimum of three times the longest pretreatment interseizure interval, or 12 months, whichever is longer. The study design and outcome does not incorporate this definition, in that the pre-treatment inter-seizure interval is not considered. Inclusion criteria are 2 prior seizures, and there is no time limit-Therefore a subject could have had 2 seizures 6 months or even a year apart, yet the outcome is time to 12 month seizure remission. At the very least, the period of time necessary to determine treatment response by the ILAE definition should be calculated for each subject, and should be provided in the outcome report. Ideally, the outcome should be time to treatment response as defined individually for each subject.

Trial team: On the patient baseline CRFs the date of the patient's first and most recent seizures are recorded. This is used to calculate the time to becoming seizure free. Unfortunately, the trial is nearly completed at this time so it is not possible to update the protocol with the IAE definition at this time.=. However it can be used to inform the trial results.

2. In the prior SANAD study, the seizure type was not taken into consideration. This is important for both focal and generalized epilepsy, and would be important for treatment providers. It would be very useful to include information of control of seizures with impaired awareness for the focal protocol, and control of GTCC for the generalized protocol. For example, if valproate and levetiracetam were equally as efficacious for treating GTCC but not myoclonus and absence, some patients might opt to use it if pregnancy was anticipated.

3. An ITT analysis is most rigorous in placebo-controlled trials, but not in comparative trials, because a subject who is quickly removed from a drug and placed on a different drug will be analysed in their original group, and side effects and seizure outcome will be attributed to the drug they were originally randomized to. This choice should be justified.

Trial team: The trial statisticians will perform both an ITT analysis and an analysis based on actual drug taken/prescribed

Reviewer 4

This is a complex one hundred pages of dense writing by a well grounded and experienced group seizure disorder specialists. Essentially, the paper is dedicated to the nature and methods to be used in a prospective trial assessing day to day clinical utility of two seizure medications.

Much of the design is based on a similar trial conducted by this group more than a decade ago. Having said, my only concern is with the possibility that the minor changes such as the duration of dependent measure may increase the risk of a type II error.

It is up to the editors of the journal as to how much trimming of the manuscript from the current 100 pages or so they want.

Trial team: Protocol paper is 19 pages long. A copy of the protocol was also provided and this is close to 100 pages long – could the reviewer have confused the 2 documents?

VERSION 2 – REVIEW

REVIEWER	rani sarkis Brigham and Woman's Hospital, Harvard Medical School, USA
REVIEW RETURNED	06-Jun-2020

GENERAL COMMENTS	concerns have been addressed.
-------------------------------

REVIEWER	Jacqueline French NYU Grossman School of Medicine I receive NYU salary support for consulting work and/or attending Scientific Advisory Boards on behalf of the Epilepsy Study Consortium for UCB, the manufacturer of levetiracetam
REVIEW RETURNED	14-Jun-2020

GENERAL COMMENTS	I am unsure how much room there is for change at this point, as the trial has already been completed. There does not seem to have been any change to the manuscript as a consequence of the comments
--

	from the reviewers. This may be because the statistical plan cannot be changed at this point, which is understandable. However certain aspects could be included in study weaknesses, such as the point that adequate dosing may not be achieved, and that ILAE definitions of treatment response have not been employed.
--	--

VERSION 2 – AUTHOR RESPONSE

Reviewer: 1

“concerns have been addressed”

Team Response:

N/A

Reviewer: 2

"I am unsure how much room there is for change at this point, as the trial has already been completed. There does not seem to have been any change to the manuscript as a consequence of the comments from the reviewers. This may be because the statistical plan cannot be changed at this point, which is understandable. However certain aspects could be included in study weaknesses, such as the point that adequate dosing may not be achieved, and that ILAE definitions of treatment response have not been employed. "

Team Response:

We have broken down the team’s response to Reviewer 2 into two parts: Response 2.1 and Response 2.2.

Team Response 2.1: “adequate dosing may not be achieved”

The protocol describes the dosing regimens for the randomised treatments. It is true that treatments may fail, usually due to adverse reactions, before an initial maintenance dose is achieved. This information will be captured in the outcome 'time to treatment failure'. We would argue that this is not a weakness of the trial design, but a strength in that meaningful outcomes are being measured. Information on dosing during the trial and treatment failure will be provided in the trial reports. We do not therefore agree that this is a weakness that should be described in the protocol.

Team Response 2.2: “The ILAE definitions of treatment response have not been employed.”

The outcomes chosen for SANAD II are those recommended by the ILAE for monotherapy trials as described in the protocol. The ILAE definition of treatment response was defined by an ILAE task whose main goal was to define treatment refractoriness. The definition of treatment response offered was not chosen as a primary or secondary outcome as it was considered inappropriate for an RCT comparing policies of starting treatment alternatives. The focus of the ILAE definition of treatment response is upon response to a specific drug rather than policy, and the definition includes achieving 'a seizure-free duration that is at least three times the longest interseizure interval prior to starting a new intervention would need to be observed.....' the aim being to provide assurance of a treatment response for an individual with a low seizure rate rather than a population in a clinical trial. This is not

a concept that is easy to explain 'at the bed side' whereas the likelihood of achieving a 12 month remission and hence a driving license is easily understood.